

# Impact of aerosols on precipitation over the Maritime Continent simulated by a convection-permitting model

Muhammad E. E. Hassim[1], W. W. Grabowski[2], and T. P. Lane[3]

[1]Centre for Climate Research Singapore, Meteorological Service Singapore, Singapore

[2]National Center for Atmospheric Research, Boulder, Colorado, USA

[3]The University of Melbourne, Melbourne, Victoria, Australia

*Correspondence to*: M. E. E. Hassim (muhammad_eeqmal_hassim@nea.gov.sg)

**Abstract.** We examine the impact of assumed cloud droplet concentration on the simulated diurnal cycle of rainfall over New Guinea and surrounding seas using convection-permitting numerical simulations with the Weather Research and Forecasting (WRF) model. The simulations mimic effects of cloud condensation nuclei on cloud and precipitation processes. They follow simulations reported in Hassim et al (ACP 2016) that focused on dynamical aspects, namely the topographic forcing and the off-shore propagation of convective systems that contribute to the observed early-morning rainfall maximum north-east of New Guinea. Simulations reported in this current study apply the bulk cloud microphysics of Thompson et al. with contrasting cloud droplet concentrations of 100 and 1,000 per cc, referred to as pristine and polluted conditions, respectively. Overall, the assumed cloud droplet concentration has a small impact on the simulated convection. This emphasizes the predominant control from the diurnal cycle and the large-scale conditions. Pristine convection results in a 15-20% larger surface accumulated rainfall over both land and ocean, and a noticeable shift of the cloud top height distribution, a reduction of the contribution of shallow cloudiness (cloud tops below 3 km) and an increase of the population of deep clouds (cloud tops above 9 km). The simulated impact on precipitation and cloud fields is in stark contrast to previous modelling studies that document small enhancement of surface precipitation and significant increase of the cloud top height in polluted conditions. Analysis of microphysical fields suggests that the simulated small enhancement of precipitation in pristine conditions comes from more efficient rain processes below the freezing level and enhanced graupel initiation and growth aloft. The increase of the cloud top height is arguably due to precipitation off-loading increasing cloud buoyancy aloft that has been shown to operate in shallow warm convection. However, the relatively low horizontal resolution and application of the bulk cloud microphysics warrants follow-up studies to assess validity of the impacts documented in the current study.



## 1 Introduction

This study is motivated by the recent attention given to the impact of atmospheric aerosols on convection, both shallow and deep, and the rainfall. Atmospheric aerosols affect the microphysics of precipitating systems through their role in nucleating cloud particles. In particular, increased concentration of cloud condensation nuclei (CCN) for a given liquid water content

leads to an increased cloud droplet concentration and thus smaller mean droplet diameter. Smaller cloud droplets delay the onset of warm-rain processes in shallow clouds through less efficient collision-coalescence, but have recently been argued to enhance precipitation in deep convection through a mechanism referred to as convective invigoration (e.g., Andreae et al., 2004; Rosenfeld et al., 2008). The invigoration is thought to originate from the enhanced latent heat release when large amounts of liquid water freeze upon being transported above the 0 degC isotherm by convective updrafts and subsequent off-

loading of frozen cloud condensate through precipitation processes. This can occur when collision-coalescence processes are suppressed in the lower portions of deep convective clouds due to high droplet concentrations and reduced droplet sizes.

A reasonable hypothesis is that polluted clouds tend to grow taller because of the invigoration. This could lead to the enhancement of ice processes, thereby increasing the surface rainfall. Satellite observational studies over continental areas

and over the Atlantic ocean seem to support this hypothesis (e.g., Lin et al., 2006; Koren et al., 2005). However, little is known as to how exactly hygroscopic aerosols activated as CCN affect mixed-phase convective clouds because freezing and riming process rates may change due to the altered overall number and size of droplets (Thompson and Eidhammer, 2014). Several observational studies have shown that the mixed-phase region is deeper in polluted environments and shallower in clean environments (e.g., Rosenfeld and Lensky 1998; Koren et al., 2005, 2008; Lin et al., 2006; Niu and Li, 2012).

However, it is difficult to separate effects of aerosols from the impact of meteorological conditions, that is, to contrast clouds and precipitation that develop in exactly the same meteorological situations and differ only in aerosols. When aerosols and meteorology vary together, the impact of aerosols may be coincidental and not causal. Moreover, the observed effects (i.e., deepening of polluted convective clouds) can be argued to originate from purely microphysical effects (i.e., more numerous and thus small ice particles having smaller terminal velocities), with an insignificant impact on the cloud dynamics

(Morrison and Grabowski 2011). Simulations with bulk single-moment microphysics reported in Grabowski (2015) and more recent simulations applying the double-moment microphysics (Grabowski and Morrison, 2016) support such a conjecture.

Numerical modelling provides a possibility of separating effects of meteorology from the impact of aerosols. In a nutshell,

one can perform simulations where meteorological conditions are kept the same, and vary only aerosols. However, one needs to distinguish between highly idealized studies (for instance, simulating only a single cloud or applying the radiative-convective quasi-equilibrium paradigm) from NWP-type convection-permitting limited-area simulations that can include realistic local meteorology. Single-cloud simulations (e.g., Teller and Levin, 2006) exclude the feedback from the cloud





dynamics and microphysics on larger-scale processes and likely overemphasize aerosol effects. This is because when a single cloud is modified by aerosols (e.g., it produces more precipitation), the cloud leaves a different footprint on its environment and thus affects the evolution of subsequent clouds. The convective-radiative quasi-equilibrium paradigm (e.g., Grabowski, 2006; Grabowski and Morrison, 2011; van den Heever et al., 2011) does include feedbacks from the cloud

dynamics on larger-scale processes, but it suffers from extremely simplified meteorological conditions. Nevertheless, convective-radiative quasi-equilibrium simulations show rather limited impacts of aerosols and highlight the role of compensating feedbacks (e.g., opposing effects in shallow and deep clouds; van den Heever et al. 2011) as well as the role of the surface energy budget that is affected by the mean size of cloud droplets and thus by aerosols (Grabowski, 2006; Grabowski and Morrison, 2011). Stevens and Feingold (2009) review various feedbacks in the context of indirect aerosol

effects and argue that these feedbacks typically mask ("buffer") the system response to aerosol perturbations.

Arguably, multiday high-resolution NWP-type simulations driven by observed meteorology provide a credible methodology to evaluate indirect aerosol effects, including the convective invigoration hypothesis. Seifert et al. (2012) were first to report such simulations applying a convection-permitting limited-area NWP model with a double-moment cloud microphysics

scheme (i.e., responding to changes of CCN and ice-nuclei, IN) for three summer seasons of convective precipitation over Germany. They note that CCN and IN assumptions have a large impact on simulated cloud properties but small impact on space- and time-averaged surface precipitation (see Fig. 9 therein). Similar conclusions were reached by Fan et al. (2013) who performed month-long NWP-type simulations over three regions (tropical western Pacific, summertime southeastern China, and summertime southern Great Plains) using the WRF (Weather Research and Forecasting) model with bin

microphysics and contrasted simulations assuming pristine and polluted conditions. The impact on the surface rain accumulation was small (below 10%) and the surface precipitation difference between pristine and polluted conditions was smaller than the difference between simulated and observationally-estimated surface rainfall. In contrast, the impact on the bulk cloud parameters (such as the cloud fraction, cloud top height, cloud thickness, etc.) was significant, in agreement with results of Seifert et al. (2012).

Following this line of thought, we consider here indirect aerosol effects on precipitation by conducting a set of large-domain convection-permitting WRF model simulations over a tropical region and contrasting convective processes in pristine and polluted environments. Our area of interest is the eastern portion of the Maritime Continent surrounding New Guinea. The study follows Hassim et al. (2016) who investigated diurnally-forced convection and associated rainfall patterns in a region

where local moist convective development is strongly tied to topographic forcing and to land-sea breeze circulations. Simulations discussed in Hassim et al. (2016) were conducted continuously for a period of two weeks (i.e., free-running) applying the nested domain model configuration. Here, we discuss results of similar simulations, but focusing on the effects of aerosols. Different aerosol conditions for each simulation are mimicked by prescribing concentrations of cloud droplets that get activated upon condensation. By simulating precipitating cloud populations under the same large-scale conditions



and varying only the cloud droplet concentration, the simulations highlight the impact of aerosols on the diurnal cycle of convection and rainfall in this region. Many clouds and cloud lifecycles are simulated, building robustness in the interpretation of results, similarly to the studies of Seifert et al. (2012) and Fan et al. (2013).

The next section describes the model and modelling setup. Section 3 presents model results focusing on the contrast between pristine and polluted simulations, and separating precipitation processes over land from those over water. A discussion of model results in section 4 concludes the paper.

## 2 Numerical simulations

### 2.1 Model configuration

Simulations in this study follow those reported in Hassim et al. (2016). They were initially performed using Advanced Research version 3.3 of the WRF model applying a two-domain configuration nested one-way. Subsequently, the simulations were repeated with the WRF model version 3.5.1, but differences between results applying the two versions of the model are small as documented in the next section. The outer domain has 540 times 420 horizontal grid points and 12-km grid spacing. The inner domain has 840 times 990 grid points and 4-km grid spacing covering New Guinea and a large

portion of Northern Australia (cf. Fig. 1a). Each domain is configured to have 80 vertical levels with a model top of 10 hPa around 30 km. The vertical grid length varies between 50 m near the surface to about 600 m at 20 km. Most of the troposphere between 4 and 14 km has the grid length of about 300 m ensuring a relatively high vertical resolution (cf. Fig. 1b). A 10-km Rayleigh sponge layer is included to absorb upward-propagating gravity waves generated by convection. To prevent the model from becoming unstable with locally large vertical velocities (particularly near steep terrain), the vertical

velocity damping option was also used. Interim re-analysis data from the European Centre for Medium-Range Weather Forecasts (ERA-Interim) were used to supply the initial, lateral and lower boundary sea surface temperature (SST) conditions for all simulation domains. The lateral boundary conditions for the outer domain and the lower boundary SST conditions for both domains were updated every 6 hours. The model's outer domain used a 30 s timestep while the inner domain was integrated applying a 10 s timestep. The first 12 hours of the simulations were considered spin-up and model

output was saved hourly thereafter for analysis. Hassim et al. (2016) and Vincent and Lane (2016) have both evaluated the performance of this model configuration, albeit with different microphysics to those used here, against satellite derived rain products, radar and gauge measurements.

Two WRF simulations were performed over a two-week period, contrasting clean and polluted atmospheric conditions. The

simulations were initialized on 1200 UTC on 1 February 2010 and end at 1200 UTC on 15 February 2010. This period was selected because the Maritime Continent region experienced suppressed large-scale conditions according to outgoing longwave radiation (OLR) data from the Year of Tropical Convection campaign (i.e., a suppressed phase of the Madden-



Julian Oscillation, MJO, over the region), thereby maximizing local diurnal forcing. The idealized pristine (PRIS) and polluted (POLL) scenarios were created by prescribing the cloud droplet concentrations to 100 cm$^{-3}$ and 1000 cm$^{-3}$, respectively. The prescribed droplet concentrations acted as proxies for different aerosol loadings in the atmosphere.

The WRF model includes many options to represent key atmospheric and surface processes. The physics packages for the land-surface, radiation, boundary-layer, and parameterized convection (the latter only for the outer domain) were selected as in Hassim et al. (2016). The 4-km inner domain was considered convection-permitting, thereby convective processes were treated explicitly. Cloud microphysical processes were parameterized with the Thompson scheme (Thompson et al., 2008). Because microphysical processes are the focus of this study, a brief review of the scheme is provided in the next section.

**2.2 The Thompson microphysics scheme**

The model simulations in the 4-km inner domain rely heavily on the details of the microphysics scheme used. The Thompson scheme (Thompson et al., 2008) is considered a 6-class bulk microphysics scheme. It explicitly predicts the mixing ratios of water vapour, cloud water, cloud ice, rain, snow and graupel. In addition, it is a double-moment scheme for cloud ice and rain, thereby predicting their number concentrations as well. Being double-moment in cloud ice and rain, the
scheme computes differential sedimentation for both species according to their mass-weighted and number-weighted terminal velocities. Concentration of cloud droplets is prescribed and it allows contrasting precipitation processes in pristine and polluted clouds as described in the previous section.

The Thompson scheme is rather unique among other bulk schemes available in WRF because it uses lookup tables to mimic
some processes usually found in more sophisticated spectral bin microphysics parameterizations. It also differs from other bulk schemes by assuming that the particle size distribution of each hydrometeor species except snow follows the gamma distribution:

$$N(D)=N_0 \, D^{\mu} \, e^{-\lambda D} \quad , \qquad (1)$$

where $N_0$, $\mu$, and $\lambda$ are the intercept, width, and slope parameters of the distribution, respectively. Setting the width parameter $\mu$ to zero reduces the expression to the classic Marshall-Palmer exponential distribution. The size distribution for snow is computed as the sum of an exponential and gamma functions, and it depends on both temperature and mixing ratio. Snow is also assumed to be non-spherical and has a variable bulk density according to a power law the links the particle mass m and its diameter D as given by $m(D)=0.069D^2$.


Cloud water in the scheme is formed with a prescribed droplet number concentration $N_c$ that gets activated upon condensation. The default value is set to 100 cm$^{-3}$, mimicking pristine maritime conditions, and it remains constant in space and time. Simply changing this value to 1000 cm$^{-3}$ allowed us to create the polluted scenario for the purposes of this study.





As cloud droplets are assumed to follow the gamma size distribution, setting a different $N_c$ value also alters the shape parameter (see Eq. A4 in Thompson et al., 2008).

The cloud ice number concentration is constrained such that the mass-weighted mean diameter is between 30 and 300 μm to prevent mass-number imbalance. However, because cloud ice is immediately transferred into the snow category once its size exceeds 200 μm, the upper size limit for cloud ice is never exceeded. Cloud ice is initiated following the Cooper (1986) curve but only after ice supersaturation exceeds 125% or the vapour mixing ratio is 100% saturated with respect to water and the temperature is below -13 degC. The heterogeneous freezing of water droplets via immersion freezing follows the Bigg (1953) formulation; larger drops are frozen into graupel and smaller droplets are frozen into cloud ice. Homogenous freezing occurs when the temperature falls below -38 degC.

Rain water in the scheme is also unique because the size distribution changes significantly depending on whether rain comes from warm-rain processes (i.e., via collision-coalescence) or ice processes (i.e., from melted ice, mainly snow and graupel), see discussion in the Appendix A in Thompson et al. (2008). Rain has a variable intercept parameter that depends on the rain mixing ratio (Eq. A6 in Thompson et al., 2008) and whether there is snow or graupel above a given level in the column. Graupel is represented with an exponential distribution with the intercept parameter that is a function of the graupel mixing ratio rather than simply a constant as in most bulk parameterization schemes. Graupel in the scheme is created primarily from rain and snow collisions. However, the collection efficiency varies as a function of the mean volume diameter of the rimed species unlike in other bulk schemes where it is often assumed constant.

As with all microphysical schemes, the Thompson scheme also undergoes refinement and modification with each new WRF model version. We repeated our pristine and polluted simulations with the Thompson scheme found in the WRF version 3.5.1 using the same model settings as described earlier. Changes to version 3.5.1 of the Thompson scheme were relatively minor. They include small modifications to the constants used to calculate the graupel slope parameter and the autoconversion of cloud droplets to rain which follows Berry and Reinhardt (1974) except for assuming a gamma distribution for the droplet spectrum. Leading coefficients for the self-collection of rain were also slightly reduced. The efficiency of rain evaporation was also reduced in the presence of melting ice (graupel/snow) such that water-coated ice evaporates slower and sheds less water when it melts. These modifications have a rather small impact because rainfall accumulations are close for the two versions of the model as documented in the next section.



# 3 Results

## 3.1 Rainfall amounts and the diurnal cycle

Figure 2 shows the two-week accumulated rainfall in the New Guinea region for PRIS and POLL cases. Simulated totals from WRF version 3.3 (left panels)) are compared to totals simulated from WRF version 3.5.1 (right panels). Overall, the

spatial distributions of the accumulated rainfall are close for different model versions, although there are some differences in details that can perhaps be attributed to different flow realizations. Notably, the domain averaged rainfall accumulations are equivalent for both PRIS and POLL simulations in both model versions, with more rainfall occurring in the PRIS scenario. Most of rainfall falls over high topography, but there is also a significant rainfall over coastal regions and the surrounding ocean. Both sets of model simulations show more rainfall in the pristine case at mostly the same locations, particularly on

the northern slopes of the New Guinea highlands and along the coastal regions. More rainfall is present further offshore in the region of (142-145$^o$E, 0-3$^o$S) in the polluted case from WRF version 3.5.1 when compared to version 3.3.

Figure 3 shows Hovmöller diagrams of the 3-hourly surface rain rate averaged over part of New Guinea and the ocean north of the island for simulations with model version 3.5.1 (top panels), together with the average diurnal cycle (middle panels).

The figure is constructed the same way as Fig. 4 in Hassim et al. (2016), that is, by averaging the surface rainfall along the island. The motivation for the figure in Hassim et al. was to compare the simulated precipitation patterns with those observed by the Tropical Rainfall Measuring Mission (TRMM) satellite during the course of many diurnal cycles. Here, we present the figure to document that precipitation develops and propagates in a similar manner in the two scenarios. This implies that large-scale conditions have the dominant effect on day-to-day variability and the effects of aerosols are relatively minor.

Because of contrasting rainfall mechanisms over land and over ocean (cf. Hassim et al. 2016 and Fig. 3 herein), subsequent analysis considers atmospheric columns over land and over ocean separately.

Figure 4 and Fig. 5 show the timeseries of area-averaged rainfall and the mean accumulation for land and water columns, respectively, and for the two model versions. The figures show that both model versions provide similar results and that the

mean rainfall features a strong diurnal cycle with some variation arguably due to effects of evolving larger-scale conditions within the models. Accumulations over land are four-to-five times larger than over the ocean because of the strong topographic forcing (cf. Fig. 2). Most interestingly, however, there is more rain in the pristine case for the two model versions and the pattern is consistent throughout the 2-week period with only a few exceptions (e.g., Feb. 3 over land). By the end of the simulation, differences between pristine and polluted cases are 15 to 20% for both water and land, with the

percentage difference larger over water. This is significantly larger and of the opposite sign than the impact documented in Seifert et al. (2012) and Fan et al. (2013).





Figure 6 shows the mean diurnal cycle over the 2-week period for polluted and pristine conditions and for WRF simulations applying version 3.5.1 (results for the version 3.3 are similar and are not shown). The mean diurnal cycle over land and water in each case is expressed as hourly anomalies, first calculated with respect to the 24-hr mean accumulated rainfall each day and then averaged over the 15-day simulation period. The figure shows that the simulations reproduce well-known tropical rainfall characteristics. Over land, convective rainfall typically peaks in the early evening hours due to the strong daytime surface forcing (e.g., Guichard et al. 2004, Grabowski et al. 2006, and references therein). Over the ocean, on the other hand, diurnal cycle of convective rainfall is significantly weaker and the rainfall peaks in the early morning hours, due to effects of offshore propagating gravity waves (e.g., Hassim et al. 2016, Vincent and Lane 2016) as well as longwave radiative cooling and lack of solar radiation at night combined with small differences between daytime and nighttime SSTs, (e.g., Gray and Jacobson 1977, Randal et al. 1991). The figure also shows that there are small differences between pristine and polluted cases, but it is unclear if these are statistically significant.

Because convective and stratiform precipitation in the tropics is produced by dynamically different processes (e.g., Houze, 1997) subsequent analysis separates the rainfall into convective and stratiform for columns over land and water. The partitioning is done by applying the texture-based Steiner algorithm (Steiner et al., 1995) to the simulated radar reflectivity at 3 km, but includes slight modifications to the peakedness criterion suggested by Penide et al (2013). Figure 7 shows the time series of mean hourly rain rate (already shown in Fig. 4 and 5) over land and water separated into convective and stratiform columns for the model version 3.5.1. The figure shows that the PRIS case typically features more rainfall than the POLL case for both convective and stratiform columns. The time evolutions separated into convective and stratiform columns are noisier than the total. Note that while the ratio between mean convective and stratiform rainfall amount is about 10:1 (c.f. compare scales on vertical axes in the upper and lower panels of Fig. 7), the mean stratiform rain area over land and water is about 10 times larger (not shown). Hence, the proportion of total convective to total stratiform rainfall amount in the model is about 1:1, consistent with observational studies (e.g., Schumacher and Houze, 2003).

**3.2 Bulk cloud field properties**

The results documented in the previous section, that is, more surface rainfall in the pristine case, can perhaps be explained by the preponderance of shallow convective clouds where warm-rain processes dominate. Such an argument can be discarded by a straightforward inspection of model results that clearly shows deep convection as the dominant rain-producing cloud system over both the land and the ocean. This is arguably a realist representation of the dominant cloud systems, though admittedly the 4 km grid spacing of the model means that shallow convection is poorly resolved. Results of a more convincing analysis are shown in Figs. 8 and 9. Figure 8 compares frequency distributions of cloud-top height (CTH) for both pristine and polluted conditions, over land and over water, simulated by different model versions. CTH is calculated on a column-by-column basis, that is, a single model cloud spanning several model columns results in a set of CTHs rather than a single CTH value. CTH in each vertical column is defined as the level at which the total condensate path integrated



downwards from the model top reaches 0.01 kg m$^{-2}$. The figure clearly shows that CTH distributions are dominated by deep clouds, over both land and ocean, and for both polluted and pristine conditions. The distributions are bimodal and feature shallow clouds with tops between 2 and 3 km (slightly deeper over land), and deep clouds featuring CTHs in the upper troposphere (with the peak slightly higher over water). Most interestingly, there is a consistent change between pristine and

polluted conditions: the frequency of shallow and mid-level clouds (up to 6 km) is larger in POLL, and the frequency of deep clouds is larger in the PRIS case. Again, this is true for both model versions and for land-versus-ocean columns, and opposite of what one might expect in regards to the convective invigoration hypothesis. In fact, the CTH distribution differences between POLL and PRIS are exactly the opposite of results presented in Seifert et al. (2012; cf. Fig. 8 therein) and in Fan et al. (2013; cf. Fig. 5 therein). We do note that our full simulation and analysis domains are much larger than the

domains used by Seifert et al. and by Fan et al. in particular, allowing for better sampling of the overall modelled cloud population over a much wider region.

Figure 9 shows profiles of the mean cloud fraction for simulations using model version 3.5.1. Cloud fraction at a given height is defined as the fraction of gridpoints featuring more than 0.01 g kg$^{-1}$ from the sum of cloud water, cloud ice, snow

and graupel. Despite relatively coarse horizontal resolution, the model seems to produce realistic cloud fraction profiles with the tri-modal distribution featuring low-, mid-, and high-level clouds (peaks between 1 and 2 km, 5 and 6 km, and around 12 km, respectively) as typically observed in the tropics (e.g., Johnson et al. 1999). Although the difference between PRIS and POLL simulations are relatively small, they are consistent with the differences in the CTH distributions shown in Fig. 8. For instance, higher cloud fraction of the upper-tropospheric clouds for the PRIS case agrees with the larger contribution of these

clouds to the CTH distribution. Similarly, slightly larger cloud fractions for the POLL case between 2 and 3 km seem to agree with the CTH distributions for shallow clouds. Overall, the differences in CTH and cloud fractions are relatively small suggesting that the differences in the accumulated precipitation come mostly from microphysical effects, and not from modified cloud dynamics.

Changes in the CTH documented in Fig. 8 are reminiscent of changes documented in Wyszogrodzki et al. (2013; Fig. 13 there in particular) and concerning effects of collision-coalescence on the CTH in shallow convection. Applying idealized single-cloud simulations, Wyszogrodzki et al. argued that the shift of the distribution, that is, reducing/increasing the number of shallower/deeper clouds, comes from the impact of precipitation on cloud buoyancy. In a nutshell, allowing clouds to precipitate more readily increases cloud buoyancy and consequently extends contribution of deeper clouds. This is because

condensed water carried by the updraft provides a negative contribution to the buoyancy and letting the condensate to fall out (i.e., off-load) through precipitation processes reduces condensate loading and leads to deeper clouds. Such a mechanism can also operate in deep convective clouds, essentially offsetting the invigoration of convection through latent heating. In fact, the impact of off-loading the condensate on the buoyancy has a similar magnitude as the freezing of the condensate: the former is $\Delta q$ (in units of the gravitational acceleration; $\Delta q$ is the mixing ratio of the condensate carried by the updraft) and





the latter is $L_f \Delta q / (c_p T) \approx \Delta q$ (where $L_f$ is the latent heat of freezing, $c_p$ is the air specific heat at constant pressure, and $T$ is the air temperature).

The changes of cloud buoyancy and their effect on the vertical velocity are subtle and difficult to explain by analysis of model fields. This can be done efficiently through the piggybacking methodology (Grabowski 2014, 2015), but such an option is not yet available in the WRF model. Figure 10 shows histograms of the vertical velocity at 3, 8 and 12 km height. At 3 km, cloud updrafts are typically weaker than 13 m/s. There are more downdrafts in the PRIS case but more updrafts between 5-15 m/s in the POLL case, perhaps in agreement with slightly larger cloud fractions around this height. Cloud updrafts can reach 30 m/s in the upper troposphere and there are slightly more updrafts and downdrafts in the PRIS case at 12 km, again in agreement with higher cloud fractions for that scenario. However, the differences between the respective distributions are relatively small (especially at 8 and12 km) and beg the question of their statistical significance.

Figure 11 shows the mean diurnal evolution of the strongest convective updrafts (as given by the 99th percentile of vertical velocities larger than 1 m/s) over land and over water, and for pristine and polluted conditions. Although quite noisy, the figure shows small (and perhaps statistically insignificant) differences between pristine and polluted conditions. The figure also illustrates the differences in the diurnal cycle over land and water, with strongest updrafts in the upper troposphere in the afternoon over land, and weaker updrafts and noticeable enhancement in the late-night/early-morning hours over water.

As documented in idealized simulations discussed in Grabowski (2006), Grabowski and Morrison (2011), and Morrison and Grabowski (2013), differences in the latent heating between pristine and polluted cases can affect the mean temperature profile through the convective feedback. Such changes are quite subtle, ~1 K [see for instance Fig. 1, 4 and 5 in Morrison and Grabowski (2013)]. Nevertheless, such changes can lead to a noticeable modification of the Convective Available Potential Energy (CAPE) and thus affect convective processes. Figure 12 shows results of CAPE analysis for the pristine and polluted simulations in the same format as the middle panels of Fig. 3 (i.e., documenting the mean diurnal cycle). Over land, CAPE develops during morning hours and is destroyed by convective processes in the evening and nighttime hours. Over ocean, CAPE is significantly more uniform across the diurnal cycle, with a weak nighttime peak. Most importantly, there are only small and unlikely statistically significant differences between pristine and polluted cases. One can thus conclude that PRIS and POLL simulations differ little as far as mean atmospheric stability is concerned, further supporting the dominant effect of cloud microphysics, and not the convective dynamics, on surface precipitation as documented earlier in the paper.

## 3.3 Microphysical fields

The impact of increased cloud droplet concentrations on other microphysical fields holds the key to the relative differences in the accumulated rainfall. The impact is quantified through time-averaged, conditionally-sampled profiles separated into convective and stratiform regions. Conditional sampling of the cloud water mixing ratio includes grid points with the sum of



cloud water, ice, snow and graupel mixing ratios larger than 0.01 g kg$^{-1}$. The rain water mixing ratio and raindrop concentration are averaged over points where the rain water mixing ratio is larger than 0.001g kg$^{-1}$. Ice, snow and graupel mixing ratios are averaged over grid points where the sum of cloud water, ice, snow and graupel mixing ratios are larger than 0.001 g kg$^{-1}$. Conditionally-sampled profiles for convective and stratiform columns over land are shown in Figs. 13 and 14.

Profiles for columns over water are very similar to those over land and are not shown.

Figure 13 shows averaged profiles of the cloud and rain water mixing ratios as well as the rain drop concentrations. Cloud (rain) water mixing ratios are lower (higher) for PRIS than for POLL. This is in agreement with the notion that efficient collision-coalescence in the pristine case leads to a faster conversion of cloud water into rain. The magnitude of cloud water
profiles is similar in convective and stratiform regions, but rain is almost an order of magnitude larger in convective regions. This is consistent with almost an order of magnitude larger mean convective precipitation rate (cf. Fig. 7). The difference in the vertical extent of rain profiles between convective and stratiform regions reflects most likely different mechanisms of rain formation: conversion from cloud water within convective updrafts versus melting of snow and graupel in stratiform columns. Since there is approximately the same number of conditionally-sampled rain data points at the surface (not shown),
higher rain mixing ratio for PRIS in both convective and stratiform regions is consistent with the higher overall surface rainfall in the pristine simulation.

Figure 14 shows similar profiles for the ice fields, that is, cloud ice, snow, and graupel mixing ratios. In general, profiles of ice species are similar in convective and stratiform regions (with the exception of cloud ice), with systematic differences
between PRIS and POLL simulations. Contrasting cloud ice profiles in the middle troposphere are likely due to the differences in ice initiation between convective and stratiform regions. The convective profile in the pristine simulation features larger maximum at 11 km, but the magnitudes at other levels are quite similar (e.g., mean values at 8 and 12 km are around 1 and 3 mg kg$^{-1}$, respectively). Mean snow profiles are remarkably similar between pristine and polluted simulations in both convective and stratiform regions, but their magnitude is 2-3 times larger in the convective region. More significant
differences exist for the graupel profiles. First, the pristine simulation features larger values, most likely due to lifting and freezing of more abundant raindrops into the middle and upper troposphere in convective updrafts and their subsequent advection into stratiform regions. Note that higher rain mixing ratios in the stratiform regions for the pristine case (cf. Fig. 13) arguably come from melting of graupel that is higher in PRIS case. Second, the graupel maxima around 6 km are about an order of magnitude larger in the convective regions than in the stratiform regions as one might expect. When integrated in
the vertical, the graupel water path is significantly (50 to 100%) higher in the pristine case (not shown), and about an order of magnitude higher in convective regions. These results emphasize the role of riming and formation of graupel in the precipitation development in deep tropical convection. This is consistent with results discussed in Grabowski (2015) who showed 40 to 60% reduction of the surface rain accumulation when applying a simple cloud microphysics scheme (that excludes graupel as a separate ice category) when compared to a more comprehensive scheme featuring graupel-like ice.





Some but not all profiles shown in Figs. 13 and 14 are consistent with profiles shown in Fan et al. (2013; see Fig. S5 in Supporting Information). Cloud water/rain mixing ratio is higher/lower in polluted clouds below the freezing level here and in Fan et al. Cloud ice profiles differ little between PRIS and POLL here (perhaps with the exception of larger cloud ice mass in the pristine case within convective areas around 11 km), but are quite different in Fan et al., with larger values in polluted cases in both convective and stratiform regions. The relatively similar cloud ice profiles here can be attributed to the microphysics scheme, which transfers cloud ice exceeding 200 μm immediately into the snow category. Furthermore, the contrast between snow and graupel profiles is not as pronounced in Fan et al. as it is here, but pristine cases do show higher values in the graupel profiles in both convective and stratiform regions in the two studies. Overall, the differences may come from differences in convective system organization and evolution, but they may also come from the differences in microphysical parameterizations, namely a bulk scheme used here and a bin scheme in Fan et al. (2013) as well as from differences in the conditional sampling.

## 4 Discussion and conclusions

Simulations of convective processes over the Maritime Continent, focusing on the area of New Guinea and surrounding seas, were conducted with the WRF model in a convection-permitting configuration (horizontal grid length of 4 km) and applying a bulk microphysics scheme that allows contrasting precipitation processes in either pristine or polluted environments. The two-week-long simulations during suppressed large-scale conditions (i.e., when the diurnal cycles over land and over ocean are clearly defined) apply two nested domains and follow the study of Hassim et al. (2016). Lateral boundary conditions of the outer domain and SSTs over the two domains come from the ERA-Interim re-analysis. As argued in the introduction, such large-domain multi-day simulations provide the best setting to study effects of activating aerosols on convective precipitation. Two relatively similar versions of the WRF model were used with both providing similar results.

In contrast to the convective invigoration hypothesis, current simulations demonstrate that surface precipitation is reduced 15 to 20% in polluted conditions over both land and ocean (Figs. 4 and 5). This is similar to the well-understood suppression of warm rain in polluted environments due to reduced cloud droplet sizes and less efficient collision-coalescence. In deep tropical convective clouds simulated in this study, the same mechanism seems to dominate over the invigoration associated with the potential for additional latent heating above the freezing level and off-loading frozen condensate through precipitation processes. In the pristine case, there is more rain below and more graupel above the freezing level, the latter arguably because of rain advected in updrafts across the freezing level and serving as a source of graupel upon freezing aloft. As shown by the cloud top height analysis (Fig. 8) the polluted cloud population features relatively more shallow clouds and less deep clouds than in pristine conditions. We suggest that this might be a signature of less efficient off-loading of cloud condensate in polluted conditions (through both collision-coalescence and ice processes) that limits cloud vertical




development. This is in stark contrast to double-moment bulk simulations discussed in Seifert et al. (2012) and Grabowski and Morrison (2016) and bin microphysics simulations of Fan et al. (2013).

Seifert et al. (2012) show that convection invigoration can explain a small (arguably insignificant) increase of summertime surface precipitation over Germany in polluted conditions. Current simulations show the opposite and quite significant effect in deep tropical convection over the Maritime Continent. A possible explanation is in the depth of the warm-rain layer, that is, the distance between the cloud base and the freezing level. Over summertime midlatitude continents, the freezing level is typically about a kilometer lower than over tropical oceans, and the cloud base is often several hundred meters higher due to lower surface relative humidity. This implies that the depth of the layer in which warm-rain processes operate is significantly smaller over summertime midlatitude continents than over tropical oceans. Arguably, warm-rain processes are very efficient in tropical maritime convection in pristine conditions. Our simulations suggest that suppression of warm rain processes by the increase of CCN concentration in such clouds leads to reduction of the surface precipitation and not to the increase through convection invigoration. However, such a conjecture does not fit results discussed in Fan et al. (2013) as one of the regions considered in that study is the tropical western Pacific (TWP) warm pool and small increase of the surface precipitation and higher cloud tops in polluted conditions were simulated. Arguably, the TWP analysis domain used by Fan et al. is much smaller than ours (~10-15°S, 128-134°E) and encompasses much flatter topography (c.f. Fig. 1) that possibly favours the development of certain organized cloud systems over others. The presence of steep orography and more complex terrain in our larger simulation/analysis domain potentially give rise to a much more varied cloud population over land due to terrain-induced mesoscale circulations. Interestingly, May et al. (2011) observed high number concentrations of smaller rain droplets in convective storm cores during low aerosol conditions (and vice-versa) when comparing a large sample of island-based (Hector) thunderstorms over the Tiwi Islands, located in the same region modelled by Fan et al. They surmised that in the low aerosol regime larger supercooled droplets get lofted high up within the updrafts. The bigger frozen drops then splinter into large numbers of small ice crystals, grow and eventually produce the higher rain number concentrations observed. Our results (Fig. 13) are similar to their findings, although the Thompson microphysics scheme used in our study does not parameterize the Hallet-Mossop ice multiplication mechanism invoked.

It is also possible that current simulations misrepresent the overall effect because of a relatively low horizontal resolution combined with the application of bulk microphysics. Bulk microphysics represents sedimentation of precipitation particles through the mass-averaged terminal velocity. If the fall velocity has a smaller magnitude than the updraft velocity, than the entire precipitation field is lifted. In contrast, because larger precipitation particles fall with higher fall velocities, differential vertical transport of the precipitation field is possible with bin microphysics. This especially concerns the rain field because the transport of drizzle and rain into the middle and upper troposphere strongly affects graupel formation and thus the development of precipitation in deep convection. In addition, low spatial resolution likely enhances the difference between bulk and bin microphysics due to less spatial variability of the cloud vertical velocity. Low spatial resolution is also relevant





for the microphysics-dynamics interactions (e.g., Bryan and Morrison 2012), the key element of the convective invigoration hypothesis. As documented in Hassim et al. (2016), simulated topographically-forced convection over New Guinea features significantly larger surface rainfall when compared to (admittedly quite uncertain) satellite observations. It is quite possible that low-resolution convection simulations overestimate surface precipitation, for instance, because of excessively wide

clouds and thus low entrainment and difficulty representing the transition from shallow to deep convection. Microphysics-dynamics interactions may change significantly when higher spatial resolution is applied (e.g., Bryan et al. 2003, Bryan and Morrison 2012). It follows that follow-up simulations, applying higher spatial resolution and different microphysical schemes are needed to further support conclusions from the initial set of simulations presented in this paper. Microphysical piggybacking methodology (Grabowski 2014, 2015) can also be used to assess the impact with better confidence. We hope

to report on such simulations in the future.

Acknowledgments

WWG was partially supported by the DOE ASR Grant DE-SC0008648. NCAR is sponsored by the NSF. TPL is supported by the Australian Research Council's Centres of Excellence Scheme (CE110001028). High-performance computing was provided by the National Computational Infrastructure (NCI) facility at the Australian National University. Data analysis and visualisation were conducted with the NCAR Command Language (Version 6.2.1), Software, Boulder, Colorado:

UCAR/NCAR/CISL/VETS, doi:10.5065/D6WD3XH5, 2014.

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



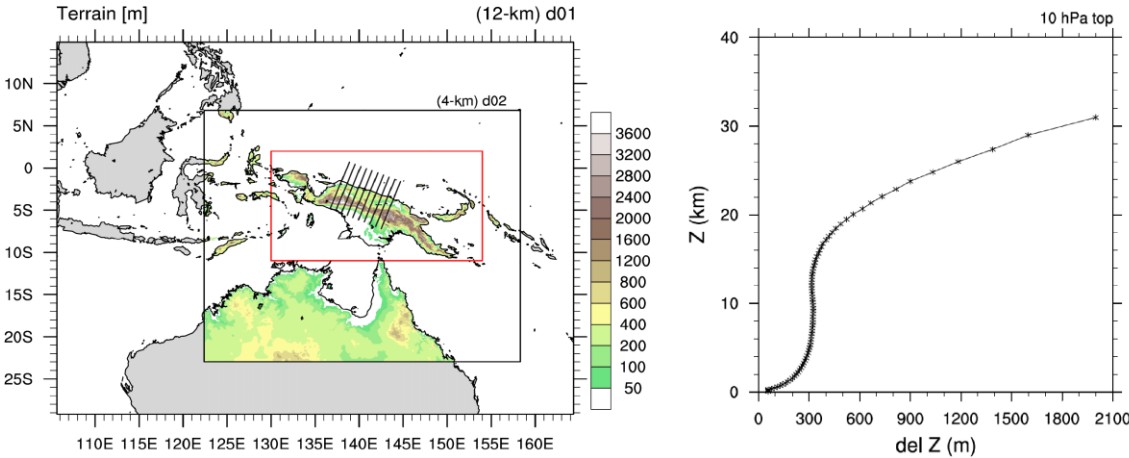

**Figure 1: Setup of WRF simulations. Left panel: outer and inner domains with horizontal grid lengths of 12 and 4 km, respectively. Model topography is shown for the inner domain using the colour scale. The line sections represent transects used to average the rainfall along the island (c.f. Fig. 3). The red box denotes the main region of analysis. Right panel: vertical grid spacing of the model as the function of height.**

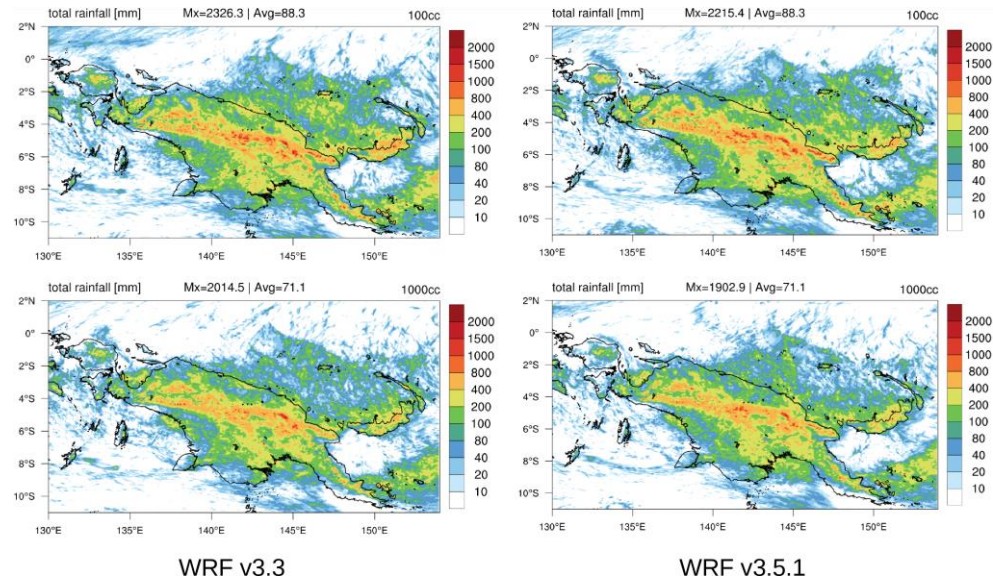

**Figure 2: Spatial distribution of the accumulated rainfall for (upper/lower panels) PRIS and POLL simulations. Left/right panels are for simualtions with WRF version 3.3/3.5.1, respectively.**





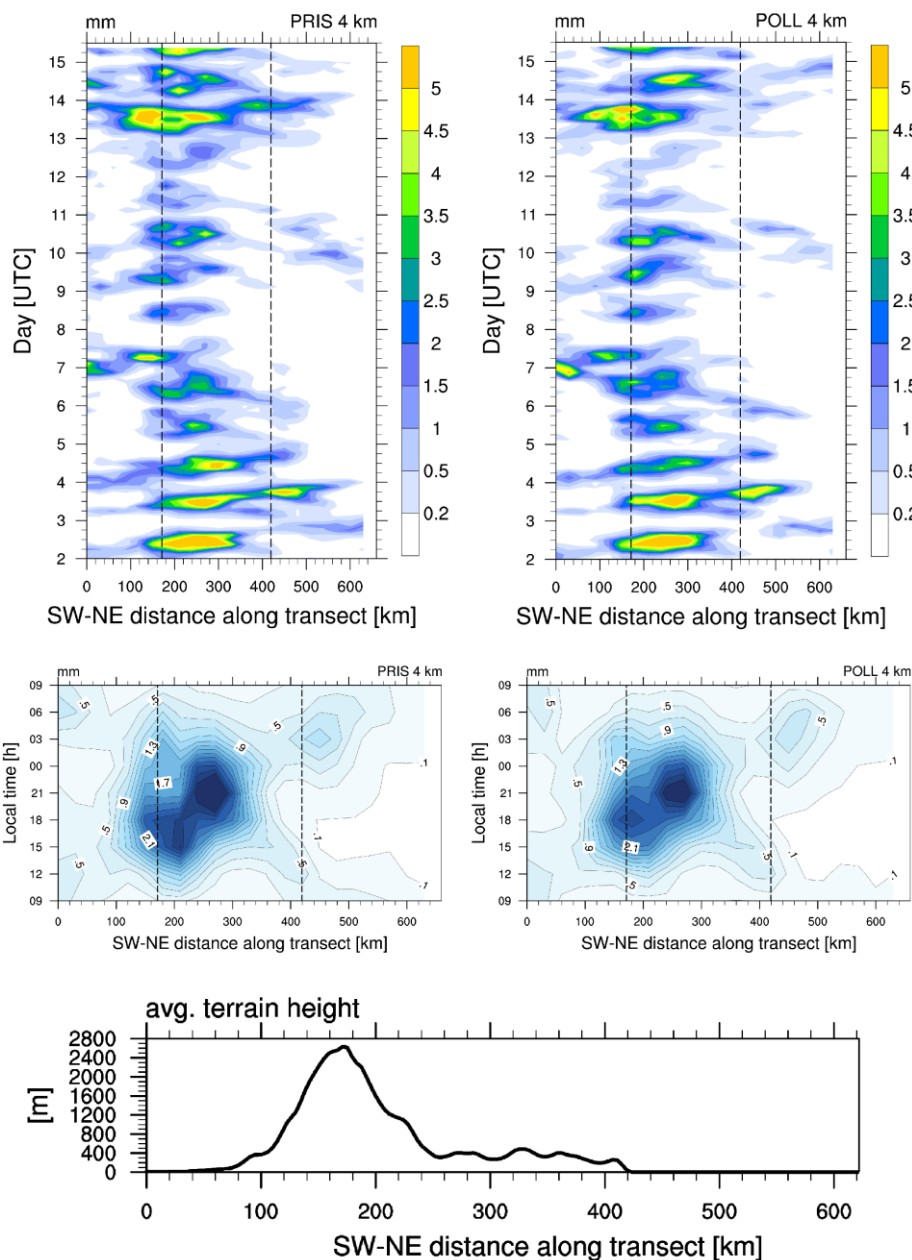

**Figure 3: Time-distance plots of 3-hourly mean rainfall (mm), averaged across the line sections shown in Fig. 1, for PRIS and**
5 **POLL simulations (top panels). The mean diurnal cycle in local time for both simulations are shown in the middel panels. The**
**averaged terrain profile is depicted by the bottom panel.**





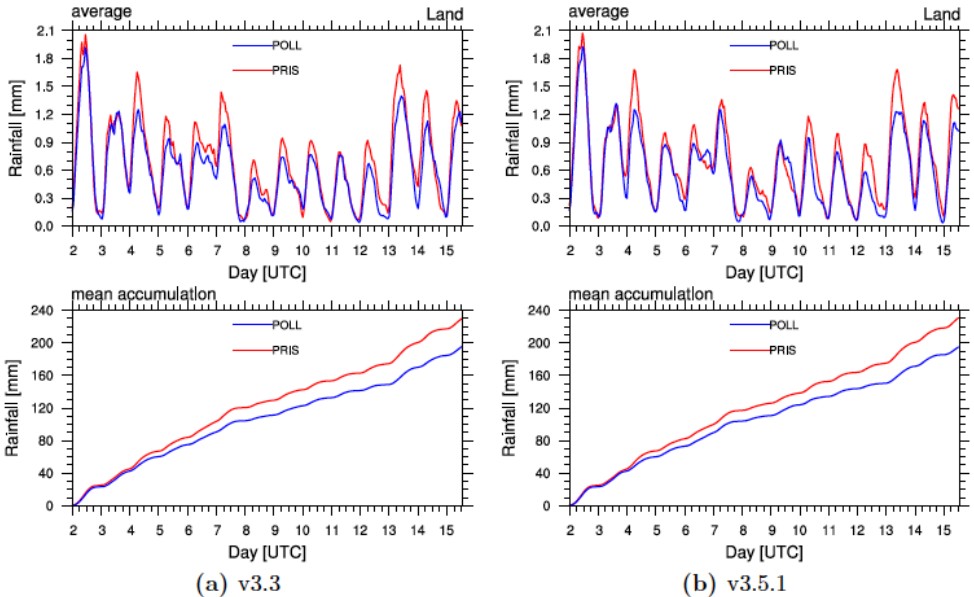

**Figure 4: Time series of hourly (top) and cumulative rainfall amount (bottom) averaged over land points in the analysis domain of Fig. 2, for PRIS and POLL simulations, in WRF model (a) v3.3 and (b) v3.5.1.**

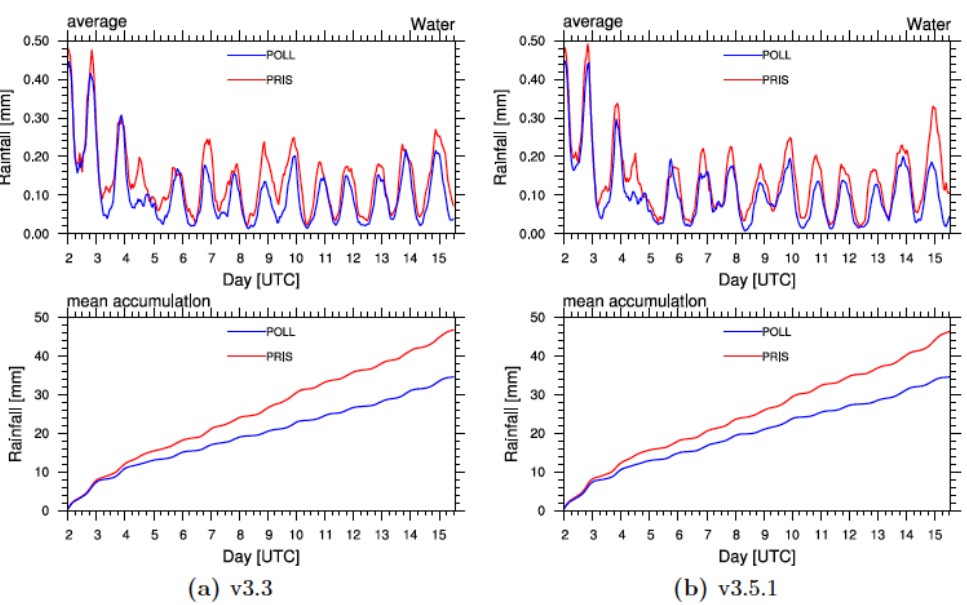

**Figure 5: As in Fig. 4, but averaged over water points.**





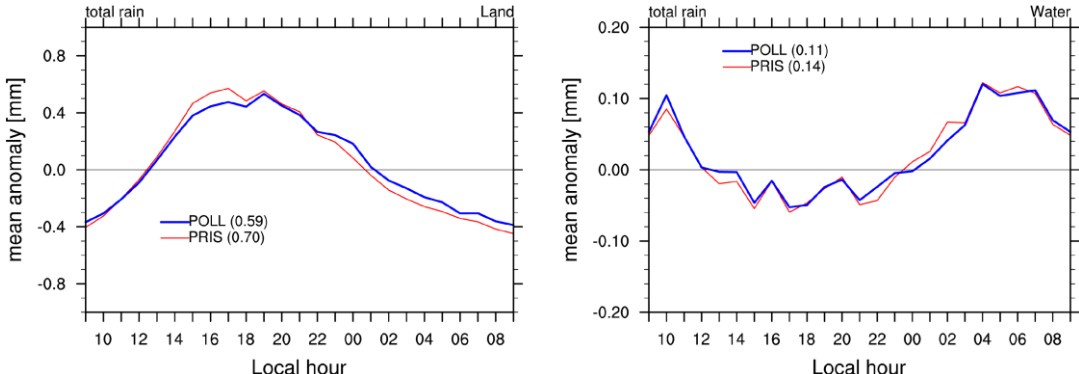

**Figure 6: Mean diurnal evolution of rainfall anomalies averaged over land (left) and water points (right), respectively, in the PRIS and POLL simulations (WRF v3.5.1). The hourly anomalies are calculated from the respective daily means, which are then averaged over the two-week period. The mean daily rainfall anomalies are shown in brackets.**

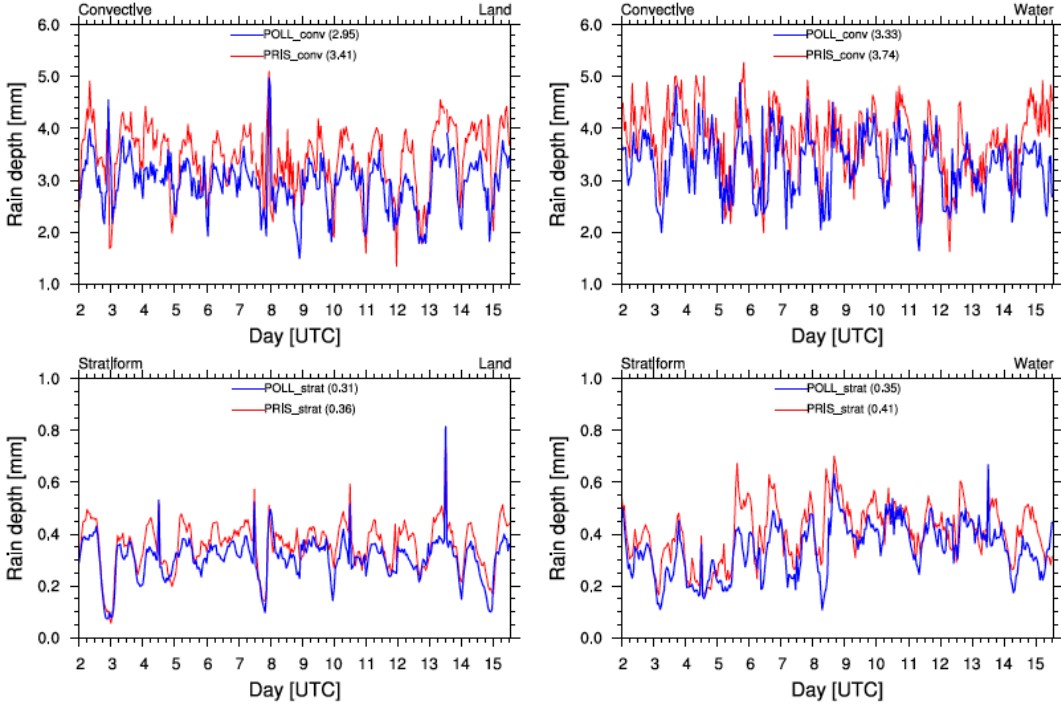

**Figure 7: Time series of hourly convective and stratiform rainfall amount, averaged over land and water for the PRIS and POLL simulations (WRF v3.5.1). The corresponding time-mean values are shown in brackets.**





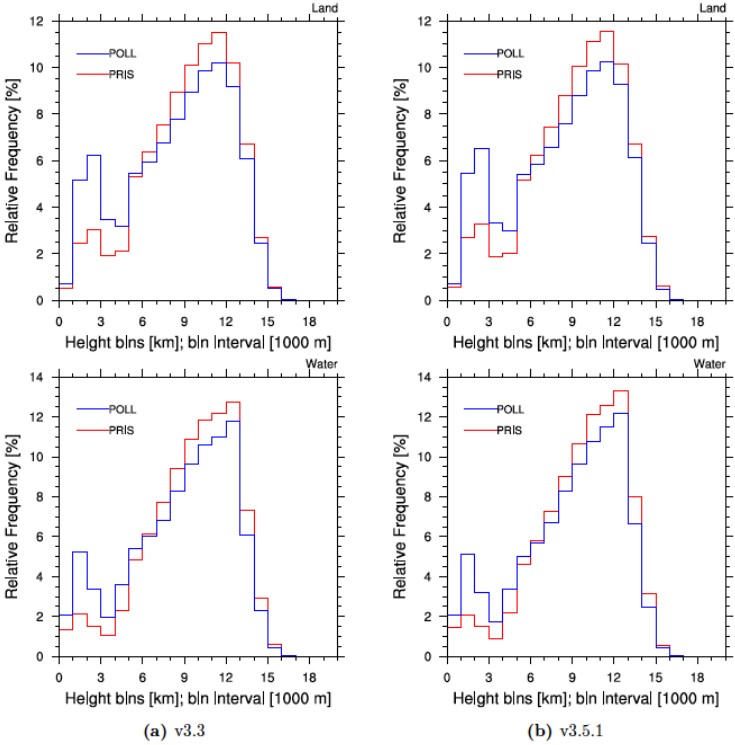

**Figure 8: Frequency distributions of cloud-top height over land and over water for PRIS and POLL simulations, from WRF model (a) v3.3 and (b) v.3.5.1.**

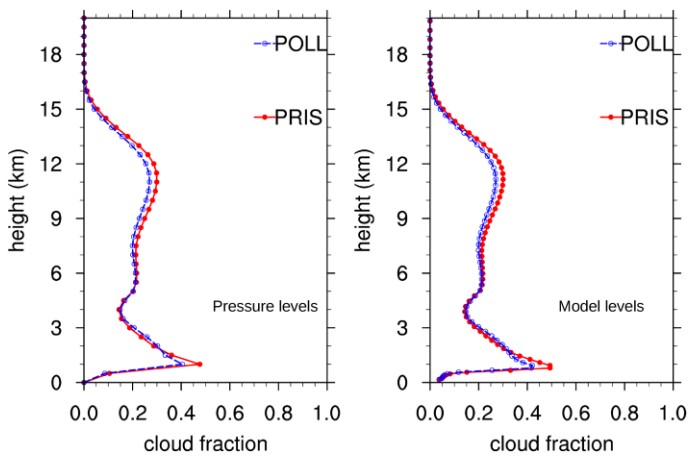

**Figure 9: Mean daily cloud fraction profiles for the pristine (PRIS) and polluted (POLL) simulations. Cloud fraction at a given height is defined as the fraction of grid points with cloud condensate (QCLOUD+QICE+QSNOW+QGRAUP) larger than 0.01 g kg$^{-1}$. Profiles are shown for (left) the data interpolated to constant height levels (every 0.5 km) and (right) data at original model levels (in equivalent height coordinates).**



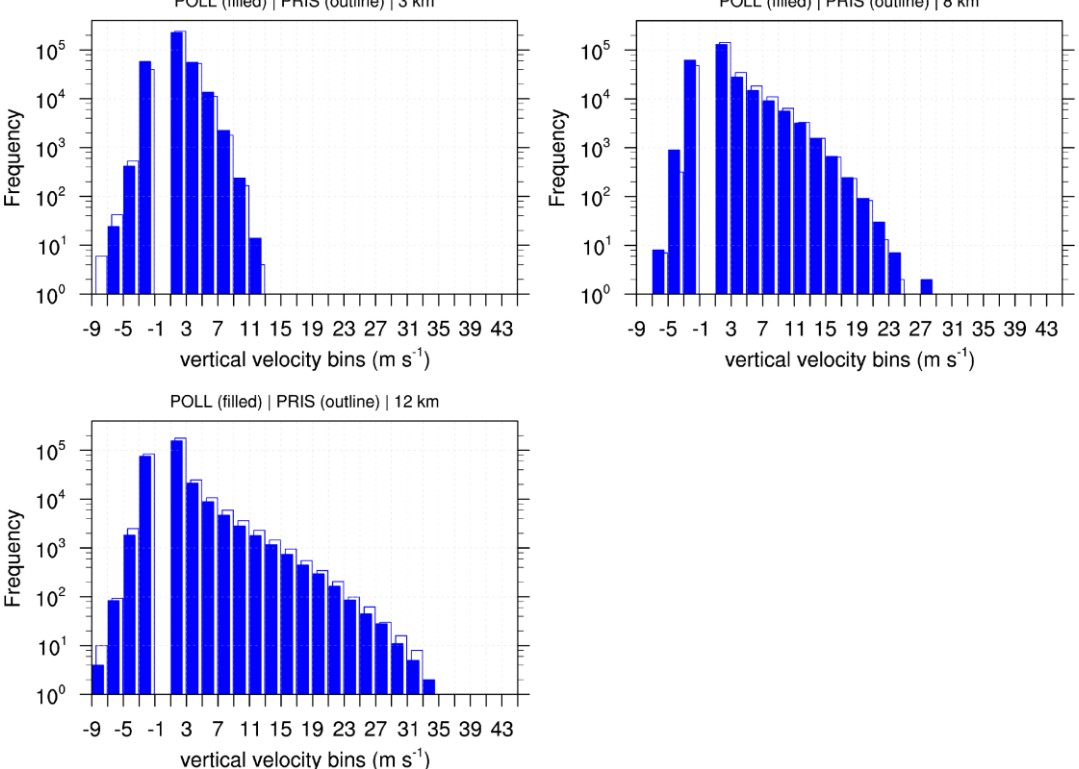

**Figure 10: Histograms of the vertical velocities for PRIS (outlined bars) and POLL (filled bars) at various height levels: (top) 3 km, (middle) 8 km, and (bottom) 12 km using hourly data. Bin width is 2 ms⁻¹. The bin corresponding to velocities between -1 ms⁻¹ and 1 ms⁻¹ has been omitted.**



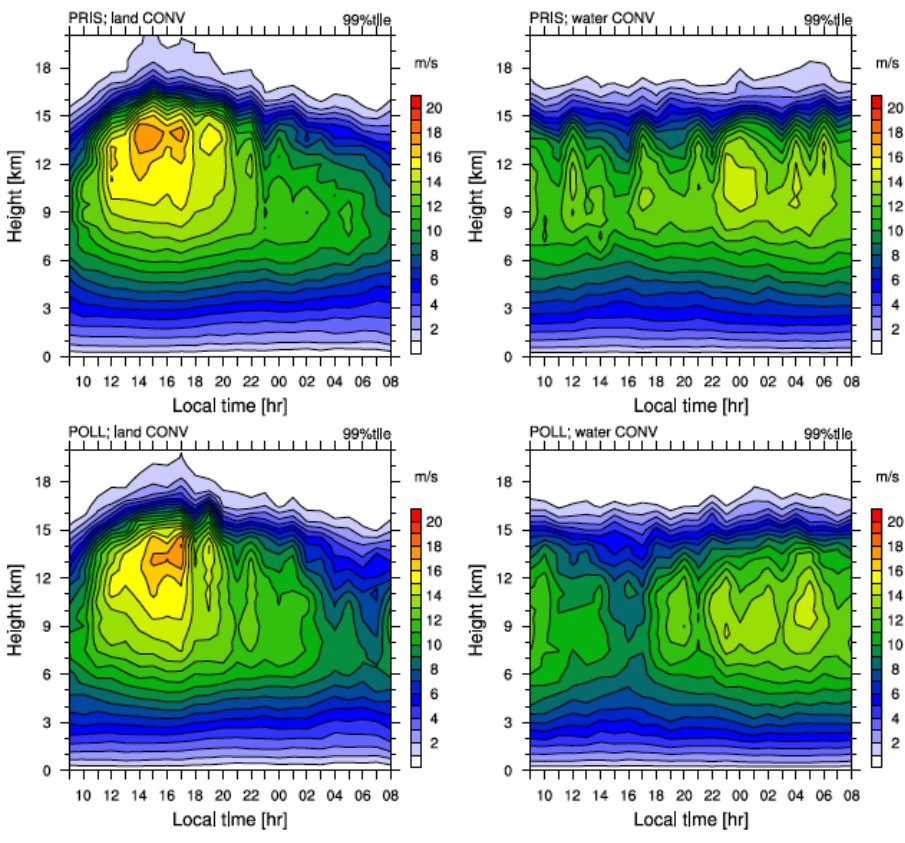

**Figure 11: The mean diurnal cycle of the 99[th] percentile of vertical velocities, averaged over land and water convective columns, respectively, for the PRIS and POLL simulations.**

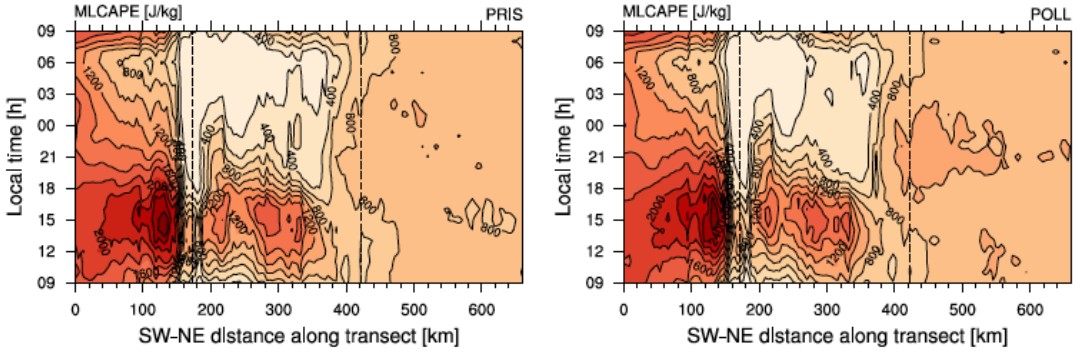

**Figure 12: The mean diurnal cycle of mixed-layer CAPE (MLCAPE), averaged across the line sections in Fig. 1, for the PRIS and POLL simulations.**





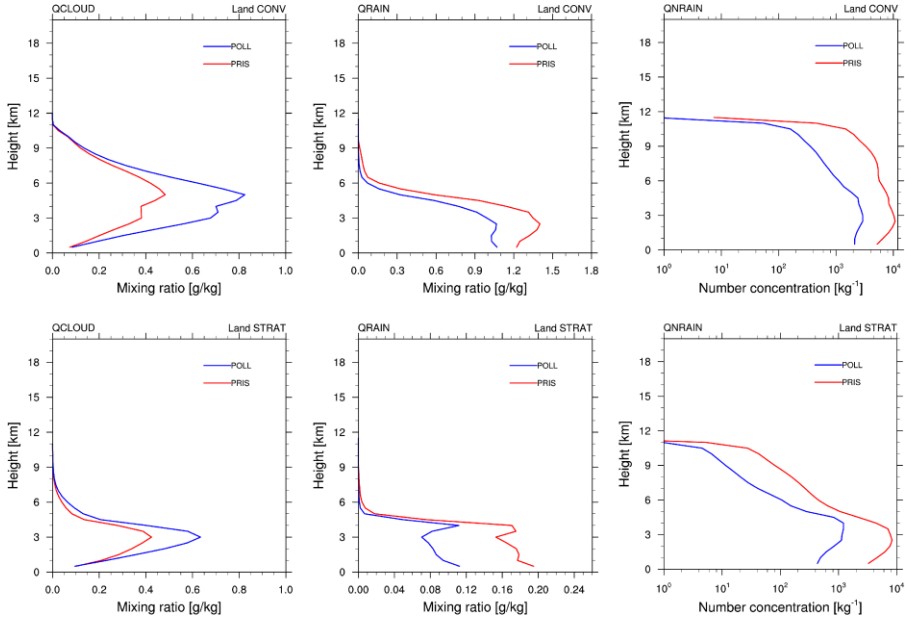

**Figure 13: Time-mean mixing ratios of cloud (QCLOUD), rain (QRAIN) and rain drop number concentrations (QNRAIN), averaged over convective (top panels) and stratiform columns (bottom panels) over land, respectively, for the PRIS and POLL simulations. The mean profiles are created after conditionally-sampling the respective microphysical fields with respect to specified thresholds (see Sec. 3.3 text for details).**

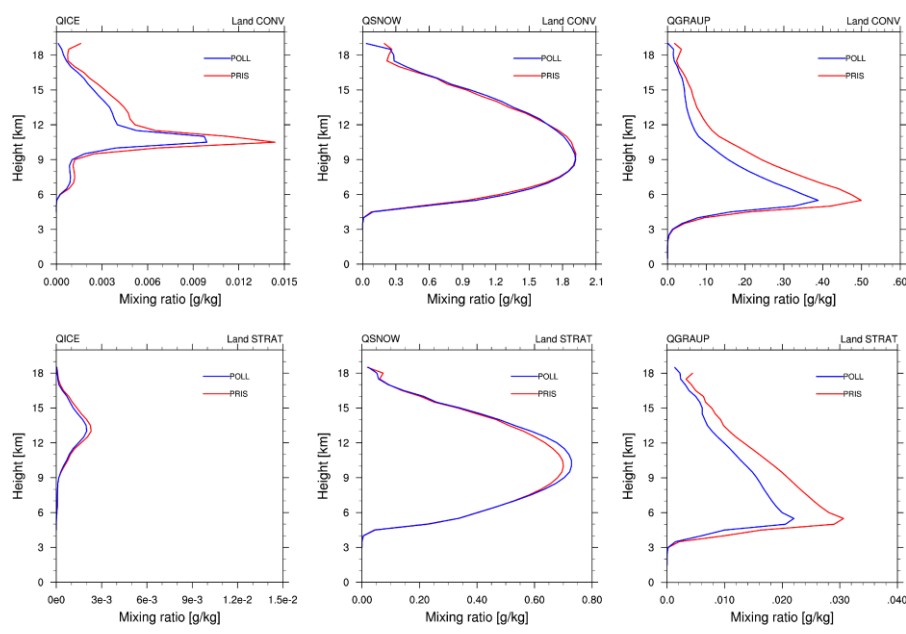

**Figure 14: As in Fig. 13, but for ice (QICE), snow (QSNOW) and graupel (QGRAUP) mixing ratios.**