# Peer review of "Impact of aerosols on precipitation over the Maritime Continent simulated by a convection-permitting model"

_Atmospheric Chemistry and Physics, 2016_

## Referee Comment (RC1) · Anonymous Referee #1 · 23 Jun 2016

In this study, the authors examined how aerosols might affect clouds and precipitation over the eastern portion of the Maritime Continent surrounding New Guinea by conducting a set of large-domain convection-permitting WRF model simulations with a bulk cloud microphysics scheme. The effects of aerosols were mimicked by contrasting model simulations with cloud droplet number concentrations of 1000/cm3 and 100/cm3, respectively. The authors found that high cloud droplet number concentrations suppress surface precipitation, with a 15-20% decrease in accumulated surface precipitation in the high cloud droplet number concentration simulation. This is in strong contrast to results from several previously documented limited-domain convection-permitting simulations. Overall, the paper is well written and the effects on cloud and

precipitation from changes in cloud droplet number concentrations are documented well and are further compared with relevant literatures. I would recommend its publication after my following comments are addressed:

Major comments: The authors attributed the decrease in the accumulated surface precipitation in the polluted case to microphysical effects (see their statement in page 9, line 21-23: "Overall, …. Suggesting that the differences in the accumulated precipitation come mostly from microphysical effects and not from modified cloud dynamics"). I am concerned with this argument in the paper. I would think the consistently higher precipitation over the two-week period in the pristine case must have contributions from feedbacks in cloud-scale dynamics or even large-scale dynamics due to changes in cloud microphysics. The authors did not explain how the extra precipitation is generated in the pristine case. The microphysical explanation the authors have in the paper helps to explain how cloud water converts to precipitation, but it does not help to explain how cloud condensate in the pristine case increases in the first place. More in-depth analysis on why the pristine case generates more cloud condensate should shed more insights on differences between this study and several previous studies.

The paper will also benefit from more in-depth analysis of microphysical processes, which might provide further insights on the discrepancy between this study and some other studies. For example, the authors noted the difference in cloud ice between this study and Fan et al. (2013). How about cloud ice number in this study? If I recall correctly, the large difference in cloud ice between pristine and polluted environments in Fan et al. (2013) are attributed to difference in cloud ice number concentrations, which leads to the difference in ice settling velocity.

More specific comments: Page 5, line 28: "a power law the links…" → "a power law that links…"?

Page 6, lines 4-10: cloud ice nucleation treatment. How about the homogeneous freezing of cloud liquid droplets? Does the number concentration of cloud droplets

affect the number concentration of cloud ice crystals? This may have implication for the discrepancy in cloud ice between this study and Fan et al. (2013).

Page 8, lines 12-16: Readers may benefit from some further clarification on how exactly the partition of convective and stratiform columens are performed.

Figure 9: The pristine simulation seems to produce consistently higher both low and high clouds. Also, Why was QRAIN not included to define cloud condensate here (see the caption of Figure 9)?

Section 3.3, the first paragraph: So cloud condensate shown in Figure 13 and 14 are in-cloud values, rather than grid-mean values, as only cloud condensate over cloudy regions are averaged?

Section 4, page 13, lines 26-33: As for the concern of bulk scheme, Seifert et al. (2012) and Grabowski and Morrison (2016) also sued bulk schemes in their studies, but their results are different from this study.

---

## Referee Comment (RC2) · Anonymous Referee #2 · 26 Jun 2016

General comment:

This study examines aerosol effects on tropical convection and the associated rainfall over the region centered at New Guinea using numerical simulations with the WRF model. The results show a small impact of differing cloud droplet number concentration on convection, and the impact is found to be of opposite sign to what is referred to as "convective invigoration". This study is of interest to the community in which the notion of convective invigoration is controversial. I would recommend this paper to be published in Atmos. Chem. Phys. after the authors address my concerns described below.

Major comments:

[Figure]

My major concern is that the authors' microphysical analysis is not enough to identify the mechanism responsible for the suppression of rainfall from convective clouds in polluted conditions. The authors invoke the classical notion of the aerosol indirect effect on warm clouds, which accounts for reduced particle size and less efficient collision-coalescence process, but it is not clear how such a microphysical modification in warm clouds influences the subsequent ice processes (including riming) that lead to the graupel formation. The authors should add comparisons of particle size and the process rate of conversion among water species (e.g. auto-conversion and riming) between pristine and polluted conditions for stratiform and convective clouds. Such an analysis would clearly demonstrate that (i) the polluted condition suppresses the warm rain process through the classical second indirect effect mechanism, (ii) the ice crystals (cloud ice in this simulation) produced from the smaller-sized cloud droplets tend to have smaller particle sizes, and (iii) such smaller ice particles have less efficiency of riming that produces graupel. If the authors add these analyses, then their findings would be substantially strengthened.

Minor points:

1. Summary of model configuration is necessary (Section 2.1). The authors state that the physics packages were selected as in Hassim et al. (2016) (page 5, line 5-7). The authors should describe the model configuration in this paper as well in a concise manner. Would it be possible to include a table that summarizes the physics schemes/packages that are employed in this study?

2. More clarified description of microphysics scheme is necessary (Section 2.2). The authors mention that cloud ice and rain are double moment – how about other species? Are they all treated as single moment (i.e. predicting only mixing ratios)?

3. There are some editorial errors. Page 5, line 28: power law the links -> power law that links Page 13, line 29: than -> then

4. The figures should be better labeled. Figure 2: Please put "PRIS" and "POLL" to

the left of the figures. Figure 7: Please put "Convective" and "Stratiform" to the left of the figures and "Land" and "Ocean" above the figures. Figures 13 and 14: Please put "Convective" and "Stratiform" to the left of the figures and "Qcloud", "Qrain", "QNrain", "Qice", "Qsnow" and "Qgraup" above the figures.

---

## Author Comment (AC1) · 22 Nov 2016

**Response to Referee 1 – Authors' response in bold**

**The authors would like to thank the Referee for reviewing our paper and for the comments provided.**

In this study, the authors examined how aerosols might affect clouds and precipitation over the eastern portion of the Maritime Continent surrounding New Guinea by conducting a set of large-domain convection-permitting WRF model simulations with a bulk cloud microphysics scheme. The effects of aerosols were mimicked by contrasting model simulations with cloud droplet number concentrations of 1000/cm3 and

100/cm3, respectively. The authors found that high cloud droplet number concentrations suppress surface precipitation, with a 15-20% decrease in accumulated surface precipitation in the high cloud droplet number concentration simulation. This is in strong contrast to results from several previously documented limited-domain convection-permitting simulations. Overall, the paper is well written and the effects on cloud and precipitation from changes in cloud droplet number concentrations are documented well and are further compared with relevant literatures. I would recommend its publication after my following comments are addressed:

***Major comments:*** The authors attributed the decrease in the accumulated surface precipitation in the polluted case to microphysical effects (see their statement in page 9, line 21-23: "Overall, . . . Suggesting that the differences in the accumulated precipitation come mostly from microphysical effects and not from modified cloud dynamics"). I am concerned with this argument in the paper. I would think the consistently higher precipitation over the two-week period in the pristine case must have contributions from feedbacks in cloud-scale dynamics or even large-scale dynamics due to changes in cloud microphysics. The authors did not explain how the extra precipitation is generated in the pristine case. The microphysical explanation the authors have in the paper helps to explain how cloud water converts to precipitation, but it does not help to explain how cloud condensate in the pristine case increases in the first place. More in-depth analysis on why the pristine case generates more cloud condensate should shed more insights on differences between this study and several previous studies.

**The Reviewer's main concern here is similar to that expressed by Referee 2, that is, where is the extra precipitation in the pristine case coming from? However, the Reviewer here is mainly concerned not with the microphysics, but rather with the large-scale water budget. We admit that we did not check the water budget in the WRF model. Ensuring conservation of the water substance is so fundamental that we expect that this was considered by WRF developers, before the model was released to the community. Because PRIS and POLL simulations are driven**

**by the same inflow boundary conditions, the difference in the surface precipitation has to come from inside the computational domain. The difference between the two cases amounts to about 17 mm of rain accumulation in 16 days over the entire domain (see Fig. 1 below). This amounts to about 1 mm/day of rain over the entire domain. The difference has to come from the difference in the surface latent heat flux combined with the difference in the outflow from the computational domain. Since 1 mm/day of surface rain corresponds to about 30 W/m$^2$, such a difference should be noticeable if the difference in rain comes from the surface latent heat flux alone. We looked at the differences in the evolution of the surface latent heat flux and the differences are about an order of magnitude smaller, a mere few W/m$^2$. It follows that the difference has to come mostly from the outflow out of the domain, i.e., drier air leaving the domain in the PRIS case. However, a detailed analysis of the outflow characteristics to support such an argument is not possible from infrequent history tapes. We included a discussion in the revised manuscript along the above points and refer to the discussion of a similar issue in Grabowski (JAS 2015) who contrasted rain accumulations and atmospheric water vapour in simulations that applied two different microphysics schemes.**

The paper will also benefit from more in-depth analysis of microphysical processes, which might provide further insights on the discrepancy between this study and some other studies. For example, the authors noted the difference in cloud ice between this study and Fan et al. (2013). How about cloud ice number in this study? If I recall correctly, the large difference in cloud ice between pristine and polluted environments in Fan et al. (2013) are attributed to difference in cloud ice number concentrations, which leads to the difference in ice settling velocity.

**Here the Reviewer's concern is similar to that expressed by Referee 2. As stated in our responses to the Referee 2 comments, it is impossible to re-run the simulations to extend analysis of cloud microphysics. An important difference be-**

tween our simulations and simulations in Fan et al. is that Fan et al. used bin microphysics. The bulk microphysics we use is only second-moment for rain and cloud ice, and only mixing ratios are predicted for snow and graupel. The similarities between cloud fractions in PRIS and POLL (Fig. 9 in the paper) are consistent to similarities in the profiles in simulations applying single-moment schemes in Grabowski (JAS 2015). However, Grabowski and Morrison (JAS 2016) show large differences in the upper-tropospheric cloud fractions between pristine and polluted conditions when using fully second-moment scheme of Morrison and Grabowski. We expanded the discussion in the revised manuscript along these lines.

*More specific comments:* Page 5, line 28: "a power law the links . . ." → "a power law that links. . ."?

**Typographical error amended in text.**

Page 6, lines 4-10: cloud ice nucleation treatment. How about the homogeneous freezing of cloud liquid droplets? Does the number concentration of cloud droplets affect the number concentration of cloud ice crystals? This may have implication for the discrepancy in cloud ice between this study and Fan et al. (2013).

**Homogeneous freezing of cloud liquid droplets occurs at temperatures below -38degC (reflected in the amended sentence).**

Page 8, lines 12-16: Readers may benefit from some further clarification on how exactly the partition of convective and stratiform columns is performed.

**Text summarising the partitioning algorithm has been added in the text for the readers' benefit.**

Figure 9: The pristine simulation seems to produce consistently higher both low and high clouds. Also, Why was QRAIN not included to define cloud condensate here (see the caption of Figure 9)?

**QRAIN is now included to define cloud condensate. This was an oversight. Figure 9 has been updated to reflect this. The pristine simulation now shows lower mean cloud fraction amounts below 5 km, consistent with the cloud-top height distribution in Fig. 8 and reflecting the more efficient raining out in pristine warm clouds.**

Section 3.3, the first paragraph: So cloud condensate shown in Figure 13 and 14 are in-cloud values, rather than grid-mean values, as only cloud condensate over cloudy regions are averaged?

**Yes, only in-cloud values were used. Showing such conditionally averaged fields documents the mean in-cloud values of various microphysical properties, but the disadvantage is that averaging is independent of the cloud fraction. Horizontal averaging, on the other hand, combines information concerning both cloud fraction and in-cloud values, and thus is less informative in our view. Following this comment, we derived horizontally-averaged profiles. Although they are different in some details from the conditionally-averaged profiles, they convey similar message as far as the differences between PRIS and POLL are concerned. We added a comment on that to the revised manuscript.**

Section 4, page 13, lines 26-33: As for the concern of bulk scheme, Seifert et al. (2012) and Grabowski and Morrison (2016) also used bulk schemes in their studies, but their results are different from this study.

**Yes, we agree. However, Grabowski (JAS 2015) applied 1-moment bulk schemes and showed results consistent with current results (more rain in the pristine case and small impact on the cloud fraction profiles). We believe the issue is related to the difference between 1-moment scheme on one side and 2-moment or bin schemes on the other side. Please note that the Thompson scheme is only partially 2-moment. We modify the discussion of model results to better expose this aspect.**

**Reference: Grabowski, W., 2015: Untangling Microphysical Impacts on Deep Convection Applying a Novel Modeling Methodology. J. Atmos. Sci., 72, 2446–2464, doi: 10.1175/JAS-D-14-0307.1.**

[Figure]

**Accumulated rainfall (area-averaged)**

POLL$_{Tot}$ (71.06)

PRIS$_{Tot}$ (88.28)

[mm]

February [UTC]

**Fig. 1.** Accumulated rainfall for the entire inner computational domain without separating land and ocean points as done in the manuscript.

[Figure]

---

## Author Comment (AC2) · 22 Nov 2016

**Responses to Referee 2 – Authors' response in bold**

**The authors would like to thank the Referee for reviewing our paper and for the comments provided.**

*General comment:* This study examines aerosol effects on tropical convection and the associated rainfall over the region centered at New Guinea using numerical simulations with the WRF model. The results show a small impact of differing cloud droplet number concentration on convection, and the impact is found to be of opposite sign to what is referred to as "convective invigoration". This study is of interest to the community in

which the notion of convective invigoration is controversial. I would recommend this paper to be published in Atmos. Chem. Phys. after the authors address my concerns described below.

*Major comments:*

My major concern is that the authors' microphysical analysis is not enough to identify the mechanism responsible for the suppression of rainfall from convective clouds in polluted conditions. The authors invoke the classical notion of the aerosol indirect effect on warm clouds, which accounts for reduced particle size and less efficient collision-coalescence process, but it is not clear how such a microphysical modification in warm clouds influences the subsequent ice processes (including riming) that lead to the graupel formation. The authors should add comparisons of particle size and the process rate of conversion among water species (e.g. auto-conversion and riming) between pristine and polluted conditions for stratiform and convective clouds. Such an analysis would clearly demonstrate that (i) the polluted condition suppresses the warm rain process through the classical second indirect effect mechanism, (ii) the ice crystals (cloud ice in this simulation) produced from the smaller-sized cloud droplets tend to have smaller particle sizes, and (iii) such smaller ice particles have less efficiency of riming that produces graupel. If the authors add these analyses, then their findings would be substantially strengthened.

**Unfortunately, it is not possible to repeat the simulations and retrieve the conversion process rates for the water species. Instead, we present in Figure 1 below the time evolution of the column-integrated water paths for cloud water, rain, cloud ice, snow, and graupel, averaged over the analysis domain, to get a sense of where the precipitation in the two cases is coming from. These diagrams and their discussion have also been added to the manuscript to supplement the averaged mixing ratio profiles already shown in the manuscript. The diagrams show that the most significant differences between PRIS and POLL come from warm-rain processes. Differences in the ice species are less signif-**

icant percentage-wise (note different vertical scale in all panels) and the PRIS minus POLL occasionally changes sign for snow. However, the difference between the 16-day mean snow paths for PRIS and POLL are similar in magnitude to the differences in cloud water and rain (0.01-0.02 g m$^{-2}$). We also expanded the discussion of other recent modeling studies.

*Minor points:*

1. Summary of model configuration is necessary (Section 2.1). The authors state that the physics packages were selected as in Hassim et al. (2016) (page 5, line 5-7). The authors should describe the model configuration in this paper as well in a concise manner. Would it be possible to include a table that summarizes the physics schemes/packages that are employed in this study?

**A table summarising physics scheme/packages used in the model study has been included.**

2. More clarified description of microphysics scheme is necessary (Section 2.2). The authors mention that cloud ice and rain are double moment – how about other species? Are they all treated as single moment (i.e. predicting only mixing ratios)?

**The Thompson scheme is double-moment only for cloud ice and rain. All other species are single-moment, with the cloud droplet concentration prescribed to 100 per cc in PRIS and 1,000 per cc in POLL. These points are now made clearer in the text (Sec. 2.2)**

3. There are some editorial errors. Page 5, line 28: power law the links → power law that links Page 13, line 29: than → then

**Errors corrected.**

4. The figures should be better labeled. Figure 2: Please put "PRIS" and "POLL" to the left of the figures. Figure 7: Please put "Convective" and "Stratiform" to the left of the figures and "Land" and "Ocean" above the figures. Figures 13 and 14: Please put

"Convective" and "Stratiform" to the left of the figures and "Qcloud", "Qrain", "QNrain", "Qice", "Qsnow" and "Qgraup" above the figures.

**Done as suggested.**

**Fig. 1.** Domain-averaged paths of various microphysical species from POLL (blue lines) and PRIS (red lines) as a function of time.